# Probiotics as Alternatives to Antibiotics for the Prevention and Control of Necrotic Enteritis in Chickens

**DOI:** 10.3390/pathogens11060692

**Published:** 2022-06-16

**Authors:** Raveendra R. Kulkarni, Carissa Gaghan, Kaitlin Gorrell, Shayan Sharif, Khaled Taha-Abdelaziz

**Affiliations:** 1Department of Population Health and Pathobiology, College of Veterinary Medicine, North Carolina State University, 1060 William Moore Dr., Raleigh, NC 27606, USA; cegaghan@ncsu.edu (C.G.); kgorrel@ncsu.edu (K.G.); 2Department of Pathobiology, Ontario Veterinary College, University of Guelph, 50 Stone Rd E, Guelph, ON N1G 2W1, Canada; shayan@uoguelph.ca; 3Department of Animal and Veterinary Sciences, Clemson University, Clemson, SC 29634, USA; khaled2@clemson.edu

**Keywords:** necrotic enteritis, *Clostridium perfringens*, probiotics, disease control, lactobacilli, alternatives to antibiotics

## Abstract

Necrotic enteritis (NE) in poultry is an economically important disease caused by *Clostridium perfringens* type A bacteria. A global trend on restricting the use of antibiotics as feed supplements in food animal production has caused a spike in the NE incidences in chickens, particularly in broiler populations. Amongst several non-antibiotic strategies for NE control tried so far, probiotics seem to offer promising avenues. The current review focuses on studies that have evaluated probiotic effects on *C. perfringens* growth and NE development. Several probiotic species, including *Lactobacillus*, *Enterococcus*, *Bacillus*, and *Bacteroides* bacteria as well as some yeast species have been tested in chickens against *C. perfringens* and NE development. These findings have shown to improve bird performance, reduce *C. perfringens* colonization and NE-associated pathology. The underlying probiotic mechanisms of NE control suggest that probiotics can help maintain a healthy gut microbial balance by modifying its composition, improve mucosal integrity by upregulating expression of tight-junction proteins, and modulate immune responses by downregulating expression of inflammatory cytokines. Collectively, these studies indicate that probiotics can offer a promising platform for NE control and that more investigations are needed to study whether these experimental probiotics can effectively prevent NE in commercial poultry operational settings.

## 1. Introduction

Necrotic enteritis (NE) is a multifactorial disease caused by toxin-producing, virulent strains of *C. perfringens* type A anaerobes and to a lesser extent by type C strains [1]. NE is of utmost importance to the poultry industry as the economic losses to the global poultry industry due to NE are estimated to be six billion USD per year [2]. As an opportunistic pathogen, *C. perfringens* colonizes the intestinal tract of healthy chickens and its proliferation is associated with production of various toxins, including α-toxin, NetB, TpeL, and perhaps, other undefined toxins which eventually cause NE [3]. There are a number of predisposing factors that enhance *C. perfringens* proliferation and toxin production, including dietary composition, immunosuppression, and intestinal damage caused by other diseases, particularly coccidiosis [4]. NE is commonly seen in 2–5 week-old broiler chickens and the clinical form of the disease is characterized by diarrhoea and high mortality rate up to 50% and thus, incurring huge economic losses to the poultry industry. The subclinical NE that often goes unnoticed incurs financial losses due to poor bird performance as a result of subtle epithelial damage leading to impaired nutrient absorption [5]. A field study that evaluated the economic impact of NE showed that broiler flocks reporting higher incidence of NE resulted in impaired feed conversion ratio (FCR) (44%), loss in live weight (27%), loss from condemnation at slaughter (19%), and mortality (8%), leading to a net loss of one third of the potential profit [6]. Another study that evaluated an impact of NE, specifically the subclinical infection in broiler chickens, reported a 12% reduction in body weight gain, and a 10.9% increase in FCR [7]. Therefore, NE affected flocks can see mortality losses of up to 30% in clinical cases resulting in financial losses, the subclinical NE characterized by reduced bird performance (weight gain, FCR, and carcass yield) can further cause substantial loss of production, and therefore loss of revenue even without high mortality rates [8].

For many decades, the use of sub-therapeutic doses of antimicrobials in feed, as growth promoters in food animal production, had kept NE under control. However, in an effort to stem the growing antimicrobial resistance (AMR) in bacteria, there has been a regulatory ban in the United States, Europe, and many other nations on the use of medically important antibiotics as feed supplements. This has consequently led to an increased incidence of NE in poultry, particularly in broiler flocks reared without antibiotics [9]. This has necessitated an urgent need for research-based efforts to design and develop effective NE control measures. Many means of NE control such as the use of acidifiers, plant-based products, prebiotics, probiotics, and vaccines have been tested both in academic and industrial settings with varying degree of success. Amongst these tested strategies, a great deal of research supports probiotics as a potential replacement for antibiotics and NE control. Probiotics are defined as “live microorganisms which when administered in adequate amounts confer a health benefit to the host” [10]. Probiotics exert their beneficial effects on poultry health though different mechanisms, such as modulation of mucosal immune responses and intestinal microbiota, improving the integrity of the intestinal epithelial barrier, alteration of mucus secretion, competitive exclusion, and production of antimicrobial and immunomodulatory substances [11]. Unlike many other antimicrobial alternatives, probiotic bacterial strains can differentially modulate inflammatory response and thus, possess an ability to regulate intestinal inflammation induced by enteric pathogens. 

Several probiotic species, including those belonging to the bacterial genera of *Lactobacillus*, *Enterococcus*, *Bacillus*, and *Bacteroides* as well as some yeast species have been tested in chickens to show some beneficial effects in preventing certain enteric diseases, including NE. In addition to improving intestinal barrier functions, probiotics can reduce the risk of infection by opportunistic pathogens either directly through producing molecules with antimicrobial activities, such as bacteriocins and/or indirectly through enhancing intestinal mucosal immune response [12,13]. The present review will systematically dissect the evidence accumulated so far to highlight the important benefits of various probiotic genera and species in reducing the enteric burden of pathogenic *C. perfringens* and thus, preventing subclinical or clinical forms of NE in chickens (summarized in Figure 1). The relevant mechanisms of action mediated by probiotic microbes will also be discussed, in the context of NE prevention and control.

## 2. *Lactobacillus* Bacteria Used as Probiotics for Control of *C. perfringens*-Induced NE

Lactic acid bacteria (LAB), over the recent years, have been recognized as promising probiotic candidates for replacing infeed-antibiotics in poultry due to their unique antimicrobial properties as well as their beneficial effects on gut health and immunity. The probiotic potential of LAB is attributed to the production of antimicrobial substances such as bacteriocins, lactate, and hydrogen peroxide, and their ability to boost the immunity of the host and competitively exclude enteric pathogens by lowering the pH of the gut and competing with them for nutrients and attachment sites. As reported by Caly et al. (2015), the most researched LAB include *Lactobacillus acidophilus, L. casei, L. fermentum, L. johnsonii, L. plantarum, L. reuteri*, *L. rhamnosus, L. salivarius*, and others [14], as discussed below.

*Lactobacillus acidophilus* is a commonly used human and livestock probiotic due to its ability to survive low pH and competitively exclude pathogenic bacteria. For example, Li, et al. (2017) evaluated the effects of dietary *L. acidophilus* in regulating the intestinal microbiota of broiler chickens challenged with *C. perfringens* and observed that *L. acidophilus* increased beneficial bacterial populations while reducing the pathogen load [15]. Furthermore, the authors found that the intestinal concentrations of lactate and butyrate, both of which have antimicrobial properties, were elevated. In a subsequent study, these authors also found that *L. acidophilus* can decrease the expression of proinflammatory cytokines and chemokines, such as IL-1β, tumor necrosis factor (TNF)-like factors, interferon (IFN)-γ, and IL-8, suggesting that *L. acidophilus* possesses the ability to alleviate NE-induced inflammation [16]. Finally, the study reported improved villus morphometry (villus height, VH; crypt depth, CD, and ratio of VH and CD), and reduced NE-associated lesions and mortality [16]. We have previously shown that *L. acidophilus* can effectively augment host antibodies against experimental antigens, such as sheep RBC (SRBC) and Keyhole limpet hemocyanin (KLH) in chickens [17]. Furthermore, *L. acidophilus*, as well as *L. fermentum*, were observed to inhibit growth and α-toxin production of *C. perfringens* and to suppress the inflammatory response caused by NE infection [18]. It is of note here that several studies have shown synergetic probiotic benefits of *L. acidophilus* and *L. casei*. For example, when poultry were treated with a commercial feed additive, Primalac^®^, consisting mainly of *L. acidophilus* and *L. casei,* a reduction in *C. perfringens* intestinal colonization was observed [19]. Primalac^®^, tested in a later study in broiler chickens, was found to reduce NE-associated mortality and intestinal pathology while regulating transcription of tight junction proteins (TJPs), cytokines, and nutrient transporters, all of which are known to alleviate NE-induced damage [20]. Collectively, these studies indicate that *L. acidophilus* and *L. casei* possess probiotic properties and that these lactobacilli can be used with a good degree of success in preventing NE in chickens.

Several strains of *L. fermentum* have been experimentally used as probiotics. For example, *L. fermentum* 1.2029 has been observed to possess a good epithelial adhesion ability, better survivability in both low pH and bile environment and an ability to reduce the severity of intestinal inflammation [21]. When tested against *C. perfringens*-induced NE, *L. fermentum* 1.2029 was shown to significantly reduce the severity of NE-associated intestinal lesions and the levels of IFNγ and toll-like receptor (TLR)2 transcription, while increasing IL-10 gene expression in the ileum, indicating an immunoregulatory potential of *L. fermentum* 1.2029 in arresting or mitigating the inflammatory damage caused by *C. perfringens* [21]. *L. fermentum* strain 104R has also been shown to have anti-*C. perfringens* activity; the co-culturing of *L. fermentum* 104R with *C. perfringens* resulted in a significant reduction of *C. perfringens* cpb2 toxin transcription, which was partly attributed to the pH-lowering ability of *L. fermentum* [22]. Although the role of this toxin in NE pathogenesis is questionable [23], the fact that cpb2 was found to be involved in epithelial damage [24] suggests a beneficial role for *L. fermentum* 104R in NE.

*Lactobacillus johnsonii* is another species that exhibits a strain-specific ability to inhibit *C. perfringens*-induced NE. A previous study observed that administration of a blend of *L. johnsonii* and an organic acid to chickens improved feed efficiency, lesion score, and *C. perfringens* counts in NE-affected birds, while no significant changes in mortality or body weight gain were observed [25]. An earlier study by La Ragione, et al. (2004) showed anti-*C. perfringens* effects of *L. johnsonii* FI9785 in specific pathogen-free chicks [26]. Amongst the strains of *L. johnsonii*, the BS15 strain has been widely studied for its beneficial probiotic effects as well as its ability to prevent NE in broiler birds. Wang, et al. (2017) demonstrated that BS15 supplementation of broilers could prevent NE-induced intestinal (ileum) villus pathology while improving body weight gain and feed conversion ratio (FCR) [27]. Furthermore, chickens supplemented with *L. johnsonii* BS15 showed an improvement in the frequencies of T cell subsets and total immunoglobulin concentrations and reduced inflammatory cytokine concentrations associated with regulation of intestinal apoptosis and inflammation [27]. The same group of researchers has also found that *L. johnsonii* BS15 supplementation enhanced serum antioxidant activity, increased total immunoglobulins and cytokine (IL-2, IFN-γ) levels, in addition to increasing CD3 and CD4 T lymphocyte percentage in the peripheral blood of supplemented birds [28]. In addition to the demonstrated ability of *L. johnsonii* BS15 to modulate the chicken immune system, it has also been found to impact liver function in Cobb 500 chicks by decreasing the contents of peroxisome proliferator activated receptor γ (responsible for fatty acid storage and glucose metabolism) and adipose triglyceride lipase (first reaction of lipolysis) in adipose tissue and serum high-density lipoprotein cholesterol. In the same context, *L. johnsonii* BS15 has also been shown to suppress the expression of acetyl-CoA carboxylase (fatty acid metabolism regulation), fatty acid synthase, and sterol regulatory binding protein-1c (lipogenesis inducer), and increases the gene expression of peroxisome proliferator activated receptor α (regulates lipid metabolism) and carnitine palmitoyltransferase-1 (responsible for beta-oxidation of long chain fatty acids). In a more recent study, *L. johnsonii* BS15 was found to reduce liver inflammation by regulating the expression of pro-inflammatory mediators and genes involved in the signalling transduction pathway, such as CD80, IL1B, PIK3R5, TLR4, TLR2A, and the proto-oncogene protein, Fos [29]. Overall, these findings indicate that *L. johnsonii* BS15 supplementation reduces liver damage by improving lipid metabolism and mitigation of fatty acid synthesis and oxidation in the liver. Taken together, these results suggest that *L. johnsonii* BS15 supplementation not only modulates liver functions but also reduces liver damage in NE-challenged birds [30]. Furthermore, BS15 seems to modify the gut microbiota composition by increasing the relative abundance of beneficial bacteria, such as *Lactobacillus* spp., while simultaneously reducing harmful bacteria, such as *Streptococcus* spp. and enterobacteriaceae spp. Based on these observations, it can be concluded that *L. johnsonii*, especially strain BS15, possesses probiotic properties and anti-inflammatory abilities that alleviate intestinal and liver damage associated with NE in broilers, making it a promising alternative to antibiotics.

*Lactobacillus plantarum* has one of the largest genomes of all LAB and is a commonly used probiotic, often being used in a variety of fermented foods [31]. Following its previous successful use as a supplement for pigs, the use of *L.*
*plantarum* as a feed additive in poultry has gained considerable attention. A recently published study by Xu, et al. (2020) observed that *L. plantarum* supplementation to layer diet increased the ratio of intestinal VH to CD and MUC2 transcription in the ileal mucosa while reducing TNF-like factor, indicating that this probiotic supplementation not only enhances mucosal integrity, but also reduces intestinal inflammation [32]. In another study, the dietary supplementation of both *L. plantarum* and *L. rhamnosus* in broiler feed has been found to significantly reduce the enteric burden of *Salmonella* spp. and *C. perfringens* as well as pathogen-induced mortality [33]. However, no significant changes in body weight gain or FCR were observed, indicating that while *L. plantarum* appears to have a beneficial effect in preventing NE, it may not have much effect on the overall bird performance and further studies are needed to address this aspect of *L. plantarum* probiotic properties.

There have also been studies that systematically isolated lactobacilli from chickens and evaluated their probiotic potential against *C. perfringens* and foodborne pathogens both in vitro and *in vivo*. For example, a recent study recovered 62 *Lactobacillus* isolates from chicken feces and assessed their ability to inhibit *S.* Enteritidis as well as their ability to tolerate low pH and bile salts and produce hydrogen peroxide [34]. The authors found that *L. salivarius* and *L. brevis* possessed superior properties for diminishing the growth of *C. perfringens* and *E. coli*. Furthermore, these lactobacilli were found to colonize chicken intestines and reduce NE severity. Along similar lines, another recent study, using MALDI-TOF-MS (matrix assisted laser deionization/ionization time of flight mass spectrophotometry), recovered 90 bacterial isolates from chickens, of which 14 were lactobacilli predominantly belonging to *L. salivarius, L. johnsonii, L. crispatus*, or *L. reuteri* species [35]. When these lactobacilli were tested against multiple pathogens, the highest inhibitory actions were observed against *C. perfringens*. In two other studies, *L. plantarum* and *L. rhamnosus* were also observed to have the greatest ability to coaggregate *C. perfringens* as well as *C. difficile,* while tolerating low pH and high bile environments [36,37]. Interestingly, a previous study also demonstrated that supernatants from *Lactobacillus* cultures have *C. perfringens* growth inhibition abilities and that *L. acidophilus strain* 3D supernatants displayed the strongest inhibitory effects [38]. Furthermore, a later study by these authors showed that *L. salivarius* treatment of broiler chickens led to reduced enteric colonization by *S.* Enteritidis and *C. perfringens* [39]. To this end, we have previously tested the ability of *L. acidophilus*, *L. reuteri*, and *L. salivarius* in enhancing antigen-specific antibody- and cell-mediated immune responses in chickens to find that birds treated with *L. salivarius* had markedly increased serum antibody responses to experimental antigens [40]. Along similar lines, a previous study that used *L. plantarum* and *L. paracasei* also showed to augment immune responses in layers and broilers [41].

In addition to the aforementioned studies, which investigated the efficacy of a single or two *Lactobacillus* species against *C. perfringens*, there are many reports evaluating a cocktail of beneficial microbes, including lactobacilli in chickens. We have recently evaluated the effects of a multi-strain lactobacilli mixture (*L. salivarius, L. reuteri, L. crispatus*, and *L. johnsonii*) administered *in ovo* on the innate and adaptive immune responses in broiler chickens [42]. The results showed improved antibody-mediated immune responses and differentially modulated cytokine expression in mucosal and systemic lymphoid tissues, indicating a possible immunopotentiation activity of probiotic mixture formulations. Indeed, while a single-strain probiotic has been shown to confer health benefits and protect against several enteric pathogens, multi-strain probiotics are thought to provide additive or synergistic effects, thus conferring greater protective effects. However, there is a lack of comparative studies to confirm this hypothesis, especially in the context of NE in poultry. A study by Layton, et al. (2013) demonstrated that a commercial *Lactobacillus*-based probiotic FloraMax^®^ B-11(FM-B11) administered via drinking water was able to significantly reduce mortality and *C. perfringens*-associated NE in broilers [43]. Although they did not find any significant improvement in gross lesions in *C. perfringens*-infected broilers, an increase in body weight gain and a decrease in ileal *C. perfringens* counts were observed. Furthermore, the authors also tested the efficacy of FM-B11 under commercial broiler settings to show that it could reduce the severity of NE and mortality comparable to the amoxicillin-treated birds. Another microbial water supplement, Synbiotic^®^, containing *L. rhamnosus*, *Pediococcus acidilactici*, and *Agave tequilana fructans* was found to improve villus morphometry and reduce colonization by *S.* Typhimurium and *C. perfringens* pathogens, while increasing lactobacilli numbers [44]. Along similar lines, a microbial feed supplement, PoultryStar^®^, composed of *L. reuteri*, *Enterococcus faecium*, *Bifidobacterium animalis*, and *Pediococcus acidilactici* showed a significant reduction in the intestinal counts of *C. perfringens* and an improved body weight gain in NE-challenged broilers [45]. Furthermore, these effects were found associated with an increased IL-1 and a decreased IL-10 transcription in the ceca of birds and markedly greater amounts of anti-*C. perfringens* secretory IgA in the *C. perfringens*-infected birds compared to unchallenged controls. A previous study that tested PoultryStar^®^ in broilers also showed similar reduction in NE severity and improved production parameters [46]. Most recently, our group has also demonstrated that administration of a cocktail of probiotics, including two isolates of *L. johnsonii* and one of *L. salivarius*, *L. reuteri*, and *L. crispatus*, to NE-challenged chickens improved intestinal morphology, induced significant alteration of cytokine gene transcription in the duodenum and jejunum and increased the abundance of some beneficial bacterial phyla in the gut microbiota, including Actinobacteria, Lactobacillaceae, and Firmicutes, associated with significantly lower lesion scores [47].

Taken together, several *Lactobacillus* species have been tested in chickens to evaluate their potential in enhancing intestinal immune health as well as in preventing important enteric infections, including *C. perfringens*-induced NE. The results accumulated so far have been encouraging and highlight the potential of lactobacilli, particularly *L. johnsonii*, *L. acidophilus*, *L. reuteri*, *L. plantarum*, and *L. salivarius*, either alone or in combination, to serve as antibiotic alternative candidates. Owing to their ability to produce bacteriocins, lactate, butyric acid, and hydrogen peroxide, and to boost immunity in the host, coaggregate pathogens, and perform competitive exclusion, lactobacilli seem to be effective against *C. perfringens*-induced NE. An important observation from these studies is also that their effectiveness depends on the age and type of the birds tested, route and frequency of administration, whether used as single or multi-strain mixtures as well as the severity of NE reproduced in the experimental challenge models. Therefore, it is important to consider these factors while devising strategies to replace antibiotics in poultry production.

## 3. Non-*Lactobacillus* Beneficial Microbes as Probiotics for NE Control

In addition to *Lactobacillus*, several other commensal bacterial species have been identified and extensively studied as probiotics for the control of *C. perfringens*-induced NE in poultry. These include *Clostridium butyricum*, *Enterococcus faecium*, *Butyricicoccus pullicaecorum*, and *Bacillus* species that include *B. subtilis*, *B. licheniformis*, *B. amyloliquefaciens*, and *B. coagulans*, as well as some yeast species such as *Saccharomyces cerevisiae* and *Pichia pastoris*.

### 3.1. Clostridium butyricum

*Clostridium butyricum*, a gram-positive anaerobic spore-forming bacterium, is a widely used probiotic due to its production of short-chain fatty acids, such as butyrate and bacteriocins [48]. Several studies have assessed the probiotic efficacy of *C. butyricum* against human *C. difficile* infections using various murine models and the results showed that *C. butyricum* possesses antagonistic effects against *C. difficile* through inhibition of *C. difficile* toxin production [48]. Importantly, *C. butyricum* used as a poultry probiotic has been found to offer antagonistic effects against *C. perfringens*. For example, Huang, et al. (2019), investigated the ability of *C. butyricum* as a probiotic to arrest the development of NE in chickens through its modulatory effects on the gut microbiota, immune response, and intestinal barrier function [49]. The authors found that *C. butyricum* supplementation reduces the expression of a pro-inflammatory cytokine IL-17A, increases the expression of the anti-inflammatory cytokine IL-10, and improves the structure of intestinal epithelium through an augmented expression of the TJP Claudin-1 gene. However, while *C. butyricum* was able to reduce the abundance of *C. perfringens* in the intestinal system, it did not significantly affect intestinal lesions and did not significantly restore the shift in microbiota composition induced by *C. perfringens* infection [49]. On the contrary, a recent study evaluated the probiotic potential of another strain of *C. butyricum* (MIYAIRI 588) in chickens experimentally challenged with *C. perfringens*. The results of this study revealed that the supplementation of *C. butyricum* to chickens not only increases weight gain and FCR, but also reduces the incidence and severity of NE-induced intestinal lesions [50]. The differences observed in these studies could be attributed, in part, to the probiotic characteristics of the strain used and on the other part, to the different experimental design. Nevertheless, it seems that *C. butyricum* can be a good probiotic candidate and that the beneficial effects of these bacteria are largely associated with their immunomodulatory abilities and production of antimicrobial molecules.

### 3.2. Enterococcus faecium

*Enterococcus faecium*, a commensal gram-positive bacterium, is another probiotic that has been experimentally used in food animals [51]. A major advantage associated with *E. faecium* is that it can colonize the gut effectively and reside for a longer period of time, which aids in the regulation of gut microbiota. Due to these attributes, it remains a common probiotic used in swine; however, it has not been used widely for poultry. A recent study by Wu, et al. (2019) found that incorporation of *E. faecium* strain NCIMB 11181 in the diet of broilers challenged with *C. perfringens* resulted in an upregulated expression of the TJP Claudin-1 and control of intestinal inflammation by regulating expression of cytokines, growth factors, heat shock proteins, and TLRs, thus contributing to reduced NE lesions in the gut [51]. Furthermore, the authors observed that *E. faecium* was able to significantly improve the gut microbial composition in broilers in both healthy and *C. perfringens*-challenged birds. Although not extensively studied, it appears that the dietary supplementation of *E. faecium* in chickens can not only help to alleviate NE development, but also can exert beneficial effects in healthy flocks through their ability to regulate resident gut microflora populations, modulate inflammation, and improve mucosal integrity.

### 3.3. Butyricicoccus pullicaecorum

*Butyricicoccus pullicaecorum*, an anaerobic bacterium, is commonly isolated from caeca of broilers. Similar to *C. butyricum, B. pullicaecorum* produces butyrate which is essential for maintaining normal gut resident microbiota and host immune homeostasis [52]. While butyric acid can often be a feed-additive in broiler diets to prevent NE, *B. pullicaecorum* supplementation allows for in situ production of butyrate. In a previous study, *C. perfringens*-challenged broilers supplemented with *B. pullicaecorum* had improved feed conversion, decreased NE lesions, and reduced enteric burden of *Campylobacter*, *Enterococcus,* and *Escherichia* pathogens [53]. However, on the other hand, a reduced body weight gain was observed in males receiving *B. pullicaecorum* supplementation. To this end, the authors argue that this observation could somewhat be advantageous to birds since the slower growth of boilers can reduce the risks associated with metabolic diseases and thus, can resist NE development more effectively. While this argument sounds plausible, it will remain speculative unless more studies demonstrate similar findings.

### 3.4. Bacillus Species

*Bacillus* species are another popular class of non-*Lactobacillus* probiotics that have gathered immense attention in the research area of antimicrobial alternatives associated with some of their key advantages. Like many other probiotic bacteria, *Bacillus* species interact with gut-associated lymphoid tissue (GALT), secretion of antibacterial factors and more importantly, their ability to form spores that are resistant to high temperature which enables them to survive the poultry feed production process [54]. Furthermore, their high tolerance to gastric acid, bile, and digestive enzymes aid their passage through the GI tract [54]. However, *Bacillus* spp. are also somewhat less efficient colonizers and require repeated oral or in-feed administrations to achieve a desired probiotic effect [52]. Several species of *Bacillus* have been shown to possess anti-*C. perfringens* activities both in vitro and in vivo models. The following are the species of *Bacillus* that have been found to be beneficial for gut health and effective against *C. perfringens*.

*Bacillus subtilis* is the most commonly researched and used *Bacillus* species in the animal feed industry [55]. As a probiotic, *B. subtilis* has been shown to improve bird performance, modulate gut microbiota population and GALT functions, and effectively interfere with colonization by pathogenic microbes, including *Clostridium* species, in chickens [56]. A recent study found that dietary inclusion of *B. subtilis* DSM 32315 to broilers challenged with *C. perfringens* resulted in higher body weight, lower mortality, improved lesion scores, and significantly lower *C. perfringens* recoverability [57]. Two other studies showed that the supplementation of broilers with either *B. subtilis* B21 alone or a mixture of *B. subtilis* B21 and *B. licheniformis* B26 can promote weight gain and FCR, prevent NE-induced damage and significantly improve villus morphometry to a level comparable to those received enramycin antibiotic [58,59]. Improvement in broiler body weight gain and FCR through *B. subtilis* supplementation compared to an NE positive control group was additionally seen in Hernandez-Patlan, et al. (2019) [60]. Jayaraman, et al. (2013) previously used *B. subtilis* PB6, a commensal broiler strain in vitro *that* possessed anti-*Campylobacter* and anti-*Clostridium* properties. Furthermore, when PB6 was tested in an experimental *C. perfringens* infection chicken model, the probiotic treatment showed improved production parameters and intestinal histomorphometry [61]. The authors in a later study also found that PB6 supplemented broiler birds not only alleviated infection severity, but also improved body weight gain, FCR and villus morphology to a level comparable to those treated with avilamycin and bacitracin methylene disalicylate (BMD) antibiotics [62]. Similarly, a previous study suggested the beneficial effects of *B. subtilis* PB6 in chickens marked by its antimicrobial activity against *C. perfringens*, *C. difficile*, *Streptococcus pneumonia*, *Campylobacter jejuni*, and *Campylobacter coli* [63]. The probiotic potential of several other strains of *B. subtilis* has also been evaluated in chickens, including C-3102 [64], DSM29784 [65], and DSM 32315 [66], and was found to significantly improve performance parameters and intestinal pathology. Mechanistically, while *B. subtilis* C-3102 could increase enteric *Lactobacillus* counts, food digestibility, and gross metabolic energy, while reducing pathogen burden and ammonia emission. The DSM 32315 strain was shown to improve pathways associated with carbohydrate metabolism and reduced pathways associated with protein metabolism, which in turn dampen the negative impact of *C. perfringens* infection in broilers. Furthermore, a study by Latorre, et al. (2015) evaluated different diets that favored *C. perfringens* proliferation, such as non-starch polysaccharides, and found that administration of a mixture of *B. subtilis* and *B. amyloliquefaciens*, both of which produced cellulase and xylanase, were able to inhibit *C. perfringens* proliferation compared to untreated control [67]. Collectively, *B. subtilis* species have been widely studied for their beneficial probiotic effects against NE in chickens and the afore-mentioned evidence associated with their use can make them good probiotic candidates in NE control in poultry.

*Bacillus licheniformis* is another commonly used *Bacillus* species in both poultry and swine as a feed supplement. Its ability to secrete antimicrobial surfactant molecules, serine protease and other various enzymes makes it a suitable probiotic species against enteric pathogens, including *C. perfringens* in poultry [68]. Zhou, et al. (2016) studied the effects of *B. licheniformis* in broiler chickens challenged with *C. perfringens* to find that its supplementation was associated with improved FCR as well as reduced oxidative stress associated with NE via reducing the activity of serum catalase and glutathione peroxidase activity [69]. Supplementing the diet with this bacterium also resulted in the upregulation of genes related to catabolism in the liver specifically carnitine palmitoyltransferase-1 mitochondrial enzyme and peroxisome proliferator-activated receptor-α nuclear receptor protein, the enzymes important in maintaining lipid metabolism and fatty-acid synthesis, thus alleviating the NE-associated liver damage. Later studies revealed that pre-treating broilers with *B. licheniformis* before *C. perfringens* challenge resulted in increased gut microbial diversity, thus ameliorating *C. perfringens*-induced disruption of microbial population [70,71]. Furthermore, the usefulness of *B. licheniformis* in chicken feed has been compared with the antibiotic activities of vancomycin in preventing NE [72]. Based on these reports, the ability *B. licheniformis* to enhance the gut barrier integrity, GALT functions, promote a healthy microbiota, and produce antibacterial lysozyme and bacteriocin like molecules [71] makes it possibly a good probiotic candidate in poultry, especially for controlling *C. perfringens* infections.

*Bacillus amyloliquefaciens* has been studied in broilers as a growth promoting probiotic; however, despite its ability to produce antimicrobial enzymes such as barnase, cellulase, protease, amylase, and xylanase [73], not many studies have tested its efficacy against NE. A previous study, examined the effects of combinations of sodium butyrate, essential oils, and *B. amyloliquefaciens* spore suspension in broilers infected with *C. perfringens*, found that while sodium butyrate and essential oils could improve body weight gain, lesion score, and villus morphology, no such effects were observed for *B. amyloliquefaciens* alone [74]. Contrastingly, a recent study involving treatment of broilers challenged with both *C. perfringens* and *E. maxima* with *B. amyloliquefaciens* alone or with BMD antibiotic not only resulted in reduced *C. perfringens* counts, intestinal inflammation and lower wet litter and footpad dermatitis scores, but also improved flock uniformity and carcass yields comparable to BMD [75]. Furthermore, Tsukahara, et al. (2018) conducted a similar study to show that *B. amyloliquefaciens* supplemented to broilers challenged with *C. perfringens*, *E. tenella*, and *E. maxima* can improve FCR and increase gut-resident *Lactobacillus* counts, while reducing lesion scores and pathogen burden [76]. Overall, the results of these studies indicate that the dietary supplementation of *B. amyloliquefaciens* can be beneficial in enhancing gut health and control of certain enteric pathogens, including *C. perfringens*.

*Bacillus coagulans* is unique in its ability to produce coagulin, a bacteriocin-like inhibitor [77], which enables this bacterium to exhibit strong microbicidal ability against pathogenic bacterial species, including *C. perfringens*. An in vitro study by Kawarizadeh, et al. (2019) assessed the ability of *B. coagulans* to inhibit *C. perfringens* growth and its production of alpha toxin, an important virulence factor in NE pathogenesis [78]. The authors found that while treatment with *C. perfringens* with *B. coagulans* culture extract led to a significant reduction of spore germination, cytotoxicity and alpha-toxin-induced apoptosis of HT-29 human colon cancer cell lines, the coculture of *B. coagulans* and *C. perfringens* caused a significant reduction in the alpha-toxin gene expression. In an in vivo study, the supplementation of broiler diet with *B. coagulans* along with *B. subtilis* and *B. licheniformis* was found to improve FCR and lean meat yield, while no significant morphological changes were observed in NE pathology [79]. Although a few other studies suggest the probiotic potential of *B. coagulans* [69,80] in general, more work is needed to examine its vital efficacy in the context of NE in poultry. More recently, a study by Hernandez-Patlan, et al. (2022), evaluated the synergistic effects of three *Bacillus* species, *B. subtilis* (AM1002), *B. amyloliquefaciens* (AM0938), and *B. licheniformis* (JD17), when used in different combinations. The authors found that combination of *B. amyloliquefaciens* and *B. licheniformis* had more beneficial effects related to their enzyme activity and antibacterial ability when compared all three strains combined. These observations suggested that that while the above mentioned bacterial species are beneficial, additional research is needed to use mixtures of probiotic bacterial species to evaluate their combined synergistic effects to maximize gut health as well as antimicrobial benefits in chickens [81].

### 3.5. Yeasts

Like many probiotic bacteria, yeasts have also gained considerable attention in recent years as an alternative platform to replace antibiotic use in food animals. The preference for eukaryotic yeasts over prokaryotic bacteria is partly because of the concern related to AMR development in probiotic bacteria and also, in part due to the antimicrobial property of yeasts. For example, yeasts are known to produce mycocins and enzymes that can degrade bacterial toxins and thus, prevent pathogen colonization by competitive exclusion [14]. In poultry, the most commonly used yeast probiotics are *Saccharomyces cerevisiae* and *Pichia pastoris*. *S. cerevisiae*, a yeast commonly used in wine and bread making, is often added in poultry feeds along with other probiotic bacteria. Avi-Lution^®^, a symbiotic feed additive which contains *S. cerevisiae*, *E. faecium*, and *Bacillus* spp., was evaluated for its ability to prevent NE in broilers that were challenged with bacitracin-resistant *C. perfringens* [82]. It was observed that while Avi-Lution^®^ was unable to reduce the severity of intestinal lesions, mortality rate was decreased and growth performance and FCR were improved. A recent study formulated 24 feed supplements to examine their efficacy in broilers against *C. perfringens* and it revealed that one of the two supplements that had *S. cerevisiae* as a main constituent showed improved production performance and a decreased *C. perfringens* enteric burden [83]. Similarly, another study that used *B. subtilis* and *S. cerevisiae* feed additives in broilers challenged with *C. perfringens* found that the treated birds had significantly reduced numbers of heterophils and heterophil-to-lymphocyte ratios while they had an increased number of lymphocytes [56]. These studies indicate that *S. cerevisiae* may possess a probiotic ability, especially when supplemented with other probiotic bacteria such as *B. subtilis*. *Pichia pastoris* is often studied as a vector for production of heterologous proteins [84], however, very little has been researched for its ability to be a beneficial feed additive. Gil de los Santos, et al. (2012) evaluated *P. pastoris* and recombinant *P. pastoris* expressing *C. perfringens* alpha toxin in broilers and found that birds receiving recombinant *P. pastoris* showed enhanced seroconversion and had improved FCR and reduced intestinal lesions [85].

Furthermore, yeast cell wall (YCW) extract, composed of polysaccharides, glycophospholipids, chitin, and beta-glucans, has been used as a feed supplement in poultry due to its beneficial effects on the host, such as improving performance, promotion of a healthy gut microbiota, and modulation of mucosal immune responses [86]. A recent study evaluated the ability of *S. cerevisiae* YCW extract to control subclinical NE in broilers and the results revealed an increased performance and flock uniformity, reduced NE lesions and improved intestinal microbial composition as well as carcass yield of treated birds [87]. The supplementation of a *B. subtilis* with YCW extract to broilers challenged with *C*. *perfringens* has been shown to reduce the enteric burden of *C*. *perfringens* and *E. coli* and augmented expression of intestinal IFN- γ, IL-1 β, and IL-12 genes [15]. Taken together, accumulating research evidence suggests that certain yeast species such as *S. cerevisiae* and YSW may have beneficial effects on bird growth performance, gut microbiota composition, and the prevention of some enteric diseases, including subclinical NE. However, further research is needed to investigate the immunological mechanisms of protection conferred by these supplements and whether their combination with other probiotic bacteria such as lactobacilli would result in more desirable effects.

While *Lactobacillus* species have typically been the most widely studied and popular probiotic bacteria used as an AGP alternative, several other bacterial and yeast species have also been found fit for use as effective probiotic poultry supplements. Of these, *Bacillus* spp. have been most extensively studied in the context of NE in chickens for reasons such as their ability to dampen *C. perfringens* colonization, reduce NE-associated pathology, improve microbiota composition and enteric epithelial integrity, modulate mucosal immune responses, and above all, their ability to withstand feed pelleting temperature. Additionally, yeasts and their extracts have also begun to show some promising results for their use as probiotics in promoting overall bird performance, gut health and NE control. However, many more studies are indeed needed before any recommendation can be accurately and explicitly considered as an antibiotic replacement for NE control in poultry production.

An important facet to recognize here in relation to all the studies discussed above is that the methods used in reproducing experimental NE in chickens can largely influence the conclusion drawn about the efficacy evaluation of various probiotic species. This is because methods used for reproducing NE have often depended on the objectives of the studies, such as virulence determination of *C. perfringens* strains, studying NE predisposing factors or efficacy evaluation of vaccines, probiotics and various other feed additives. NE reproduction methods have always required intestinal predisposition induced by feeding birds with high protein and/or wheat/non-starch polysaccharide-based diets [88,89], infection with low coccidia (*Eimeria* species) doses [90] or inducing immunosuppression (Infectious Bursal Disease vaccine) [91] prior to *C. perfringens* challenge. Other factors for consideration include the virulence level of the *C. perfringens* strain, challenge preparation (example, growth medium, incubation time and inoculum dose), frequency and duration of challenge period (example, single- or 3–5-days daily inoculations), and the lesion scoring system, all of which have been reviewed in detail previously [92]. Considering each of these factors can influence the NE severity, the level of probiotic efficacy should also be determined accordingly.

## 4. Probiotic Mechanisms of NE Control

The mechanisms by which probiotics control *C. perfringens*-induced NE depend on several factors, including the species and strain of the probiotic agent, age and type of the bird, host immune status, and importantly, the severity of NE. While general mechanisms of probiotics include restoration of the perturbed microbiota, production of antimicrobial molecules, competitive exclusion of pathogens from colonization and host immune modulation, specific mechanisms in the context of NE are highlighted in Figure 2 and the details are discussed below.

### 4.1. Restoration of Gut Microbial Composition

Intestinal microbiota dysbiosis, caused by infections with enteric pathogens including *C. perfringens*, is often associated with perturbation of the gut immune homeostasis and impaired immune regulation, consequently leading to an exacerbation of an already existing intestinal inflammation [93]. To this end, recent reports indicate that probiotics can ameliorate *C. perfringens*-induced inflammatory responses and restore the NE-associated dysbiosis, thus maintaining intestinal homeostasis [15,30,56,70,87]. A recent review that evaluated 42 studies related to the efficacy of direct feed microbials (DFM) on chicken microbiota suggested that DFM supplementation can increase the populations of *Bacillus*, *Bifidobacterium*, *C. butyricum*, and *Lactobacillus* species and reduce the burden of *C. perfringens, Coliforms*, *E. coli*, *Enterococcus*, and *Salmonella* pathogens [94]. Increasing the abundance of lactobacilli in the gut is thought to play a role in maintaining a healthy gut microbiota during NE infection, since proliferation of *C. perfringens* is often associated with a reduced population of these lactobacilli, particularly *L. aviaries* [95]. In support of this, a previous study showed that the supplementation of broilers fed with *B. licheniformis* resulted in an increased abundance of *Bacillus* and *Lactobacillus* species in the cecum with a concurrent alleviation of *C. perfringens*-induced dysbiosis effects [70]. Along similar lines, another study showed a predominance of *Gamma proteobacters* in the ileum of NE-affected chickens, while treatment with *L*. *acidophilus* was shown to restore the microbial balance by increasing the populations of members of phylum Firmicutes and reducing Proteobacteria [15]. Collectively, it is reasonable to suggest that the intestinal dysbiosis caused by enteric infections, including NE, can be restored with the aid of probiotic supplementation in poultry.

### 4.2. Maintenance of Intestinal Epithelial Integrity, Mucus Production and Competitive Exclusion of Pathogens

Intestinal epithelial integrity is governed by several factors, of which the TJPs (claudin and occludin) and the mucus production play a critical role in preventing the pathogen entry across the epithelial barrier [96]. It is known that many toxin-producing pathogens, including *C. perfringens* gain entry into the intestinal mucosal surface and subsequent mucosal damage via disrupting the TJPs [97]. Growing literature shows that some probiotic species can effectively augment TJP expression as a means of maintaining mucosal integrity in healthy birds or repairing mucosal injury induced by enteric diseases, including NE in chickens. For example, while the expression of claudin-1 and 3 genes was decreased in NE-affected birds, treatment of chickens with certain probiotics such as *C. butyricum* [49] and *E. faecium* [51] upregulated claudin 3 expression, suggesting their role in restoring the epithelial integrity. Another study also showed that the commercial probiotic, Primalac^®^ composed mainly of *L. acidophilus* and *L. casei* can significantly upregulate TJP transcription in the enteric mucosa [20]. Furthermore, one common observation with probiotic supplementation of poultry linked to TJP expression and function is an improved epithelial histomorphometry that includes VH, CD, and ratio of VH and CD that allows a larger surface area for nutrient absorption. As reviewed above in the preceding sections, many probiotic species have been shown to positively influence the intestinal histomorphometry parameters in the context of experimental NE challenge. Among these probiotics, *Lactobacillus* spp. [16,32,44] and *Bacillus* spp. [58,59,61,62,74,98].

Mucus is composed of a highly glycosylated and interlinked proteins called mucin (MUC 1 and MUC2) and secreted by goblet cells [99]. It forms an important and integral part of the mucosal barrier serving as the primary site for adhesion and colonization of both commensal and pathogenic bacteria, including *C. perfringens* [100]. Probiotics have been shown to enhance mucin production which, in turn, facilitates their local colonization, thereby preventing epithelial adhesion and invasion by pathogenic bacterial species. For example, the supplementation of chickens with *L. acidophilus*, *L. fermentum*, *L. plantarum*, or *L. casei* has been shown to upregulate transcription of MUC 1 and MUC2 and subsequent mucin production by goblet cells, providing a favorable niche for probiotic bacteria to colonize, and also a protective physical barrier against of *C. perfringens* invasion [18,32,101]. Since *C. perfringens* adhesion and toxin production results in intestinal lesions leading to a reduced nutritional status, preventing *C. perfringens* from attaching to the intestinal wall allows probiotics to act as the first line of defense against *C. perfringens* induced intestinal damage.

Upon entering the gut, probiotic bacteria colonize the intestinal mucosa by adhering to mucin-binding proteins [102] and compete with enteric pathogens for intestinal niche and nutrients, in a process termed as competitive exclusion [103]. In the context of *C. perfringens* and NE in chickens, several reports suggest that probiotic species, particularly lactobacilli, may use competitive exclusion as one of their mechanisms to prevent *C. perfringens* from colonization [14,16,82]. A previous study that tested the efficacy of *L. johnsonii* FI9785 in reducing the colonization and shedding of *S. enterica* serotype Enteritidis, *E. coli* O78:K80 and *C. perfringens* in chickens found that this probiotic could significantly prevent colonization of *C. perfringens*, while no effects were observed against other enteropathogens [26]. Furthermore, the authors suggested that *L. johnsonii* FI9785 may be given to poultry as a probiotic supplement to control *C. perfringens*. Collectively, based on the available evidence, many probiotic species capable of colonizing the chicken gut can enhance gut health by means of strengthening the epithelial integrity via restoring TJP expression and mucus production as well as competitively excluding pathogens, including *C. perfringens*, from colonizing the mucosa.

### 4.3. Immune Modulation

Many researchers, including our group, have evaluated the immunomodulatory activities of different probiotics in chickens [40,42,47,104,105,106,107,108]. Important probiotic mechanisms of immunomodulation include their interaction with cellular receptors such as toll-like receptors (TLR), stimulation of cytokines, chemokines, and mucosal IgA production, all of which lead to regulation of intestinal inflammation and maintenance of homeostasis [109,110]. Several studies suggest an immunoregulatory role of probiotics in *C. perfringens*-infected chickens. For example, *L. acidophilus* and *L. plantarum* have been shown to decrease the expression of inflammatory cytokine and chemokine genes such as IL-1β, IFN-γ, and IL-8 in chickens [16,32]. It is of note here that IFN-γ, and IL-1β are cytokines involved in systemic inflammation and apoptotic cell death while IL-8 is a chemokine which induces chemotaxis and phagocytosis. Several *Lactobacillus* spp. have also been observed to increase serum antibody concentrations [17,40,41] which can impact both the innate and adaptive immune system. Studies have also observed that probiotic supplementation causes alterations in the T-lymphocyte population including CD3, CD4 [28] and CD8 [29].

In the context of NE in chickens, it is known that *C. perfringens* can induce intestinal inflammation, consequently leading to a disruption of intestinal barrier structure and increased intestinal permeability [3]. Oral administration of *L. fermentum* to chickens has been shown to reduce NE lesions in *C. perfringens*-challenged birds that was associated with increased ileal expression of TLR2 and IL-10 (anti-inflammatory cytokine) genes and decreased transcription of IFN-γ, indicating probiotic-mediated regulation of intestinal inflammation [111]. The immunological mechanisms of protection conferred by probiotics were further investigated by these authors to found that pre-treatment of intestinal epithelial cells with *L. acidophilus* and *L. fermentum* led to a reduction in *C. perfringens*-induced expression of the pro-inflammatory transcription factor nuclear factor kappa B (NF-kB) [18]. Along similar lines, Wang, et al. (2017) found that *L. johnsonii* supplementation in chickens challenged with *C. perfringens* alleviated inflammatory responses, correlated with an increased proliferation of IgA^+^ B cells, and CD4+ and CD8+ T cells in the ileum and a significant reduction in the ileal transcription of IL-8 and IFN-γ [27]. These observations indicate that lactobacilli possess immunomodulatory properties, and therefore, their supplementation can regulate both the innate and adaptive immune components in chickens and modulate immune functions towards a non-inflammatory phenotype, and thus, avoiding excessive immunopathology and restoring gut homeostasis.

### 4.4. Production of Antimicrobial Molecules

Probiotics secrete several substances that enhance their antimicrobial ability in addition to their host health benefits. These substances include bacteriocins, lactic acid, hydrogen peroxide, butyric acid, and postbiotics. The lactic acid produced by lactobacilli can in turn be converted to butyrate by butyric acid-producing probiotic bacteria, a phenomenon called cross-feeding interactions [52]. Butyric acid has been shown to reduce the incidence of subclinical NE in chickens [112]. Butyric acid produced by *C. butyricum* probiotic bacteria can promote epithelial cell differentiation and proliferation, serve as an energy source for other commensals [113], and exert antimicrobial activity against gram-positive and gram-negative bacteria [52]. Bacteriocins, the ribosomal synthesized peptides with broad-spectrum antimicrobial activity, are known to target specific pathogens without affecting commensal bacteria [114]. These peptide molecules operate via different mechanisms such as direct lysis of pathogens, cell–cell signaling molecules, and facilitating commensal colonization [115]. In the context of *C. perfringens* in chickens, the antimicrobial activities of bacteriocin-producing probiotic bacteria have been reported [116]. Postbiotics, the by-products of probiotic bacterial fermentation such as bacteriocins and organic acids, have also been known to act against pathogens, including *C. perfringens* [117]. It is reported that postbiotics produced by lactobacilli could increase body weight gain, decrease intestinal lesion scores, and help in immune response induction in *C. perfringens*-challenged broilers [117]. Investigation of the molecular mechanisms of protection revealed that these effects were mediated via the PI3K-Akt cell signaling and chemokine signaling pathway, mainly in the jejunum over the duodenum. While some feed additive products include organic acids, such as lactic acid or butyric acid, the supplementation of probiotics allows in situ production of these beneficial substances, thereby resulting in higher concentrations and increased benefits.

Taken together, the probiotic mechanisms of *C. perfringens* growth control and NE development in chickens include maintaining a healthy resident microbial composition and gut homeostasis, regulating intestinal inflammatory responses, enhancing mucosal epithelial integrity and production of antimicrobial molecules, and boosting overall gut health and productivity.

## 5. Conclusions and Future Directions

Increasing incidences of *C. perfringens*-induced NE in poultry flocks in the face of restricted use of antibiotics in poultry production warrant an urgent need for viable alternatives. Of the many strategies, probiotics seem to offer a promising platform for NE control. Many critical aspects are needed to be considered before making a choice of probiotic formulation for controlling NE in chickens. Some of them include that the probiotic species should be able to colonize the gut epithelium effectively, able to withstand gastric pH and bile salts, and be able to inhibit *C. perfringens* colonization, growth, and virulence. Additionally, the ability of probiotics as feed additives to remain viable throughout the feed processing is also critically important from practical consideration perspective. It is noteworthy here that probiotic efficacy depends on species and strain of choice, frequency and route of administration, as well as the age, breed, and type of the bird. Throughout this review, several bacterial and fungal species have been identified as strong probiotic candidates including *Lactobacillus* spp., *Bacillus* spp., *Clostridium butyricum*, *Enterococcus faecium*, *Butyricicoccus pullicaecorum*, *Saccharomyces cerevisiae*, and *Pichia pastoris*. While some, for example *B. subtilis* and *L. johnsonii*, have been the subjects of several research studies to evaluate their abilities and efficacy, others, such as *B. pullicaecorum* and *P. pastoris*, have not been extensively studied. Therefore, a suggested future direction may revolve around further researching various probiotic species and strains in order to determine which probiotics can and cannot be good antibiotic replacements in both improving poultry performance and in curtailing specifically the *C. perfringens*-induced NE.

## Figures and Tables

**Figure 1 pathogens-11-00692-f001:**
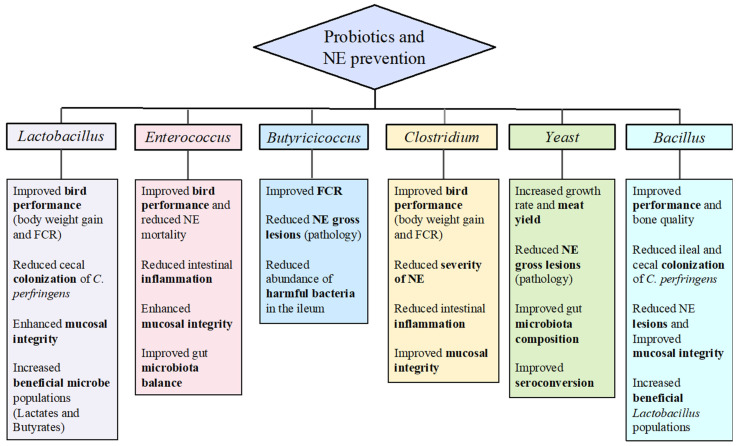
**Effects of different genera of probiotics on NE prevention**. The commonly used probiotic bacterial species in chickens for the prevention of NE belong to the *Lactobacillus*, *Enterococcus*, *Butyricicocus*, *Clostridium*, or *Bacillus* genera. Certain yeast/fungus species have also been implicated in NE prevention. The most studied experimental parameters in assessing the effects of probiotics on curtailing the NE development and progression include, measuring bird performance (weight gain and feed conversion ratio, FCR), *C. perfringens* colonization, NE lesions (gross and histopathology, including epithelial morphometry), intestinal inflammation phenotype, and microbiota composition. The beneficial effects of each bacterial genus as well as probiotic yeast are tabulated with specific NE prevention parameters.

**Figure 2 pathogens-11-00692-f002:**
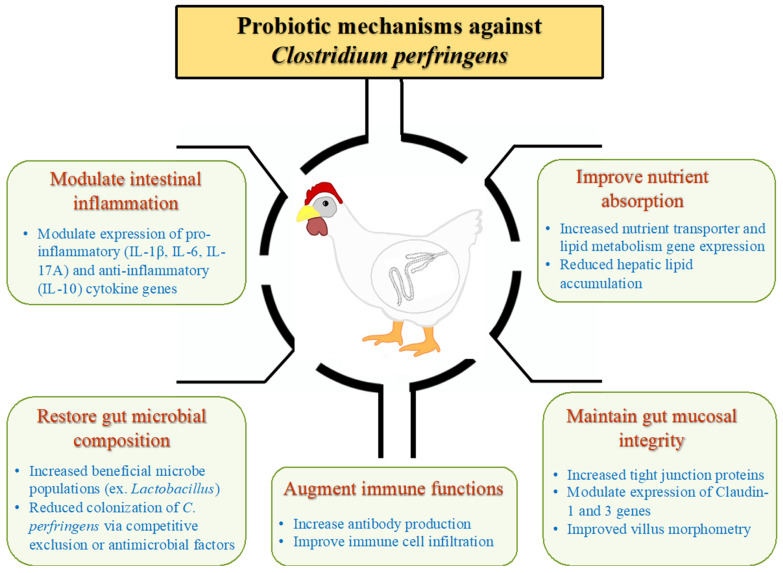
**Mechanisms used by probiotics against*****C. perfringens***. Probiotic organisms use various mechanisms to prevent *C. perfringens* colonization and NE development. Briefly, these mechanisms include: 1. Preventing intestinal inflammation by modulating the expression of pro- and/or anti-inflammatory cytokines. 2. Improving nutrient absorption via regulating the nutrient transport and lipid metabolism functions. 3. Maintaining a balanced gut microbiota population to reduce pathogen colonization. Specific mechanisms such as competitive exclusion, production of bacteriocins or other antimicrobial molecules are also involved in this probiotic function. 4. Enhancing immune functions by increasing antibody, particularly mucosal IgA, production and augmenting recruitment of macrophages, B and T cells to the mucosa. 5. Maintaining the mucosal epithelial integrity by elevating tight junction proteins expression, claudin-1 and claudin-3, as well as improving histomorphometry characteristics such as villus height (VH), crypt depth (CD) and VH/CD ratio.

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
