# Peer review of "Probiotics as Alternatives to Antibiotics for the Prevention and Control of Necrotic Enteritis in Chickens"

_pathogens, 2022, doi:10.3390/pathogens11060692_

Round 1

Reviewer 1 Report

The paper concluded the bacterial genera that prevented the NE by reducing risk of infection by opportunistic pathogens either directly through producing molecules with antimicrobial activities. But in the review, the author did not offer the effects of NE on the growth performance in chicken , and the preventing effectiveness of the various Probiotic on NE. Please provide the data.

Reviewer 2 Report

Line 215   Use apropiate writting (italics and lower case E) in S. Enteritidis

Line 242    Use apropiate writting (italics and lower case T) in S. Typhimurium, change similar situations in the whole text. 

Line 269  I suggest rephrasing "Therefore, one needs to take these factors.......production" to sound less personal.  

Line 363   Change to lower case letters when appropiate, e.g. Avilamycin, if it is not a trade name a lower case "a" is needed. 

Line 495  Figure legend is to extensive. I suugest to do a more concise writting.

Author Response

Pl see attached

Reviewer 3 Report

The information presented in the proposed review is of interest to the area of animal production, especially in poultry.

The review shows a logical sequence and although there is information that could be repetitive in different sections, the reading can be done smoothly.

To improve the review, as a suggestion, include laboratory models that can be used to induce necrotic enteritis, since a simple challenge with C. perfringens hinders the development of the disease. In addition, today's trends are the use of combinations of probiotic strains to have synergistic effects or to enhance the effect of probiotics given their different mechanisms to mitigate or control necrotic enteritis.

I enclose some references that could be useful for your knowledge

·      Impact of a Bacillus Direct-Fed Microbial on Growth Performance, Intestinal Barrier Integrity, Necrotic Enteritis Lesions, and Ileal Microbiota in Broiler Chickens Using a Laboratory Challenge Model.

·      Selection of Bacillus spp. for cellulase and xylanase production as direct-fed microbials to reduce digesta viscosity and Clostridium perfringens proliferation using an in vitro digestive model in different poultry diets.

·      Whole-Genome Sequence and Interaction Analysis in the Production of Six Enzymes From the Three Bacillus Strains Present in a Commercial Direct-Fed Microbial (Norum™) Using a Bliss Independence Test.

Finally, the file is attached with specific minimum details that must be attended to. The authors must consider that since it is a review, the writing must be done in the past tense.

Note. Check that references are complete.

Author Response

pl see attached
